# CRITIQUE TO VERIFY: ACCURATE AND HONEST TEST-TIME SCALING WITH RL-TRAINED VERIFIERS

## ABSTRACT

Test-time scaling via solution sampling and aggregation has become a key paradigm for improving the reasoning performance of Large Language Models (LLMs). While reward model selection is commonly employed in this approach, it often fails to identify minority-yet-correct answers, which limits its effectiveness beyond that of simple majority voting. We argue that this limitation stems from a lack of informative critique signals during verifier training. To bridge this gap, we introduce **Mirror-Critique**, a framework that trains a verifier with informative critiques. Our key insight is to leverage the rich critique signal by contrasting model-generated solutions with ground-truth solutions. We deploy a small instruction-tuned model to synthesize high-quality critique data with rejection sampling that teaches the verifier not only what is wrong, but also why. The synthetic data is used to cold-start the LLMs in the RLVR process to further improve the verification ability. The resulting **Mirror-Verifier** is deployed to evaluate candidate solutions by generating multiple critiques per solution, aggregating them into a verify score used for weighted voting or selective abstention. The experimental results show that our **Mirror-Verifier** significantly outperforms majority voting in terms of solution accuracy and also improves the solver's honesty to recognize and abstain from answering beyond its capability boundaries.

## 1 INTRODUCTION

Reinforcement Learning with Verifiable Reward (RLVR) has emerged as a powerful method for training Large Language Models (LLMs) to perform complex reasoning tasks, enabling significant improvements in domains such as mathematics, code generation, and scientific problem-solving. A common strategy to further boost performance is test-time scaling: generating multiple candidate solutions and aggregating them via methods such as majority voting, verifier model voting, or aggregator model selection. In an ideal scenario, an effective verifier or aggregator should be able to approach Pass@K performance by improving solution accuracy through test-time scaling. However, they often fail to identify minority yet correct solutions, resulting in limitations in their improvement compared to majority voting. This limitation underscores the need for more sophisticated verification mechanisms that can critically evaluate and select solutions.

While verifiers have demonstrated promise in detecting flawed reasoning, their training typically depends on binary labels that provide insufficient feedback about why a solution succeeds or fails. This limitation constrains the verifier's ability to improve its performance meaningfully. One potential approach involves enhancing LLMs with critique capabilities through supervised fine-tuning on critique data. However, obtaining high-quality critique data often requires sampling from closed-source models, making this approach prohibitively expensive. Additionally, the potential of RLVR for training verifiers remains largely unexplored. RLVR could offer significant advantages by improving accuracy while enabling models to recognize their limitations and appropriately abstain from answering questions beyond their capabilities.

To this end, we propose **Mirror-Critique**, a novel framework that synthesizes high quality critiques by contrasting model-generated solutions with ground-truth answers to train a verifier. Our key insight is that such informative critiques can teach the verifier not only to judge correctness, but also to understand the underlying reasoning gaps. We generate high-quality critique data via rejection sampling on an open-source, instruction-tuned model. The synthesized critique data is then used

to fine-tune the Base LLMs to address the cold start issue for the RLVR process, further improving the verification ability. The resulting **Mirror-Verifier** is deployed to generate multiple critiques per solution, which are aggregated into a verification score used for weighted voting or selective abstention during test-time scaling.

Extensive experiments on multiple mathematical reasoning benchmarks show that Mirror-Verifier significantly outperforms majority voting and reward-based selection methods, achieving superior accuracy across tasks. Furthermore, it enhances the honesty of the solver–verifier system, enabling it to abstain from questions beyond its capability boundaries, both in test-time scaling and standard (Pass@1) settings. In summary, our main contributions are:

- We introduce the **Mirror-Critique** framework, a novel approach for training verifiers that leverages rich, synthetic critique data generated by contrasting LLM solutions with ground-truth solutions. This synthetic critique data, curated via rejection sampling, provides informative signals that teach the verifier to not only identify errors but also understand their rationale. Unlike other approaches that depend on distillation from larger models, our method is self-contained, synthesizing all training data through internal supervision.

- We demonstrate significant accuracy gains in test-time scaling. By using the **Mirror-Verifier** to aggregate multiple solutions via weighted voting, our approach consistently outperforms strong baselines like majority voting and reward-model selection across multiple mathematical reasoning benchmarks.

- We propose the **honesty** score and show that Mirror-Verifier significantly improves it. This metric quantifies a model's ability to know what it knows. By abstaining from answers with low verification scores, our framework enhances model honesty while maintaining answer accuracy, reliably recognizing capability boundaries in both test-time scaling and standard (Pass@1) settings.

## 2 RELATED WORKS

**Reinforcement Learning with Verifiable Reward** Reinforcement Learning (RL) has become a standard component in the post-training stage of LLMs. Recent research indicates that RLVR substantially enhances the reasoning performance of LLMs in areas such as mathematics and code generation. A notable advancement was made with OpenAI's o1 model (Jaech et al., 2024), which marked a significant leap in reasoning capabilities. This was followed by DeepSeek-R1 (Guo et al., 2025), where RLVR was shown to activate inherent slow-thinking abilities in a base model—a paradigm now referred to as zero-RL (Li et al., 2025). Subsequently, multiple Large Reasoning Models (LRMs) have been released, such as Kimi 1.5 (Team et al., 2025), Gemini-Think (Deep-Mind, 2024), and QwQ (Qwen, 2024). SimpleRL (Zeng et al., 2025) provided comprehensive empirical studies on zero-RL, while Luo et al. (2025) utilized RLVR to further improve open-source models derived from DeepSeek-R1. A prominent RLVR algorithm adopted in many of these works is GRPO (Shao et al., 2024). Extending PPO (Schulman et al., 2017), it achieves notable improvements by evaluating multiple responses to estimate group-relative advantage. GRPO has motivated several variants, including DAPO (Yu et al., 2025), VAPO (Yue et al., 2025), and Dr. GRPO (Liu et al., 2025b). Additionally, DARS (Yang et al., 2025) introduces adaptive sampling based on difficulty, leading to gains in both Pass@1 and Pass@K metrics. In this work, we train the verifier while performing zero-RL training of the solver, and further improvements are achieved through the solver-verifier framework in test-time scaling.

**Test-Time Scaling** Test-time scaling through solution sampling and aggregation has become a widely adopted paradigm for improving reasoning performance in LLMs. A common strategy is to use rule-based methods such as majority voting, exemplified by self-consistent decoding (Wang et al., 2023; Brown et al., 2024), which aggregates multiple chain-of-thought trajectories by selecting the most frequent answer. Several lightweight variants have been proposed to enhance this approach, including dynamically adjusting the number of samples or applying heuristic filters (Aggarwal et al., 2023; Xue et al., 2023; Huang et al., 2024; Knappe et al., 2024). While effective in many cases, these methods can fail when correct solutions lie in minority modes, causing majority voting to amplify errors rather than surface the right answer. To move beyond simple counting, recent work has explored model-based selection and aggregation. These methods either train a separate reward

model to score and select candidate solutions (Yang et al., 2024b; Liu et al., 2024; 2025c), or prompt the LLM itself to compare and consolidate answers as Universal Self-Consistency (USC; Chen et al. 2024). Although these approaches combine frequency with a learned notion of quality, they can still be prone to regression errors and may not fully leverage the potential of learned aggregation. Liu et al. (2025a) propose RISE to leverage verifiable rewards from an outcome verifier to provide on-the-fly feedback for both solution generation and self-verification tasks. Concurrent to this paper, the other line of works (Qi et al., 2025; Zhao et al., 2025) explored the training of solution aggregators. Sample Set Aggregator (SSA; Qi et al. 2025), AggLM (Zhao et al., 2025) train the model aggregators via reinforcement learning to generate a final answer from multiple solutions. However, they did not utilize the informative critique information to train the model's ability to select solutions, nor did they propose a method to determine the model's reasoning boundaries to enhance its honesty. We train the LLM with synthetic critique data, guiding it to both the right answer and the exact error; this sharply improves later RLVR training. The learned verifier identifies the model's reasoning boundaries, letting it decline questions beyond its ability and greatly boosting honesty.

## 3 PROBLEM FORMULATION

We consider the problem of training a solver and a verifier from a base language model $M$ to improve reasoning performance through test-time scaling. Given a training dataset $\mathcal{D} = \{(q, a)\}$ consisting of questions $q$ and their corresponding ground-truth answers $a$, our goal is to acquire two models:

- A **solver** $S$ that, given a question $q$, generates a solution $s$ (which includes both a reasoning trace and a final answer $a$).
- A **verifier** $V$ that, given a question $q$ and a set of candidate solutions $\{s_1, s_2, \ldots, s_N\}$, selects the best solution among them.

At test time, we employ a **test-time scaling** paradigm: for a given question $q$, the solver $S$ generates $N$ candidate solutions $\{s_1, s_2, \ldots, s_N\}$. The verifier $V$ then selects the most promising answer $\hat{a}$ from the candidate set:

$$\hat{a} = f_{select}(V, q, \{s_1, s_2, \ldots, s_N\})$$

where $f_{select}$ is the selecting function that leverages the verifier $V$ to identify the solution with the highest estimated quality. The selected solution $\hat{a}$ is chosen to produce the final answer. Additionally, to enhance honesty, the selecting function $f_{select}$ can abstain from answering the questions that are beyond the reasoning capabilities of the Solver. We aim to jointly optimize the verifier $V$ such that the solver-verifier framework maximizes accuracy on the reasoning task while also improving honesty through calibrated abstention.

## 4 MIRROR-CRITIQUE FOR TEST-TIME SCALING

The Mirror-Critique framework is designed to train a high-performance verifier that leverages rich, informative critique signals. The overall framework is shown in Figure 1. This section details the four key components of our approach: (1) **RLVR Training Zero-Solver**, we use GRPO (Shao et al., 2024) to conduct RL training on the base model while collecting the trajectories generated during the training process. (2) **Mirror the Truth for Critique Synthesis**, we synthesize a large amount of high-quality critique data by contrasting model-generated solutions with ground-truth solutions; (3) **RLVR Training Zero-Verifier**, we first conduct supervised fine-tuning (SFT) to cold-start the base model, then we balance the data and conduct RL training to further improve the verifier's capabilities. Finally, we deploy the resulting solver-verifier system for accurate and honest test-time scaling.

### 4.1 MIRRORING THE TRUTH: CRITIQUE SYNTHESIS

We consider that the difficulty in training verifiers lies in the fact that relying solely on binary labels (correct or wrong) does not enable the model to understand why a solution is wrong. Critique data that points out the specific errors often requires the generation from powerful, closed-source models, which increases the cost of data synthesis. To address this issue, we propose a low-cost data synthesis pipeline that can generate high-quality, instructive critiques.

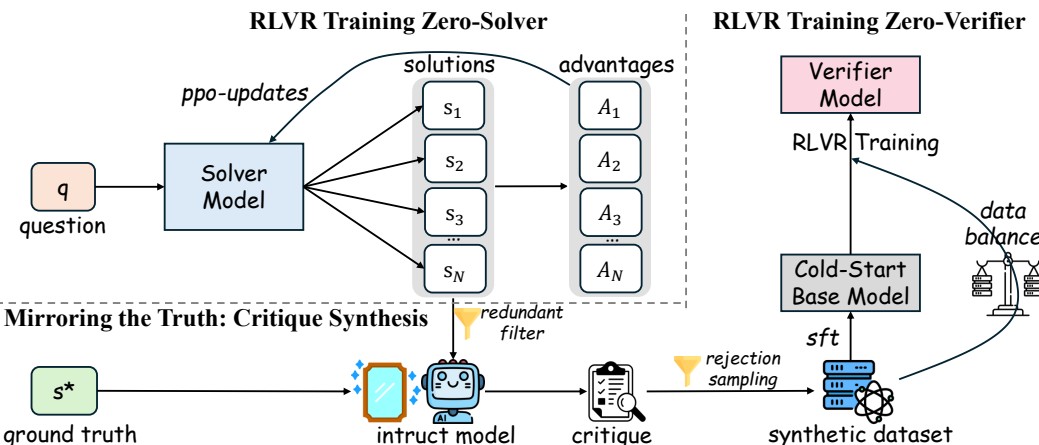

Figure 1: Overview of our framework **Mirror-Critique**. We utilized the trajectory data during the zero-solver training process of RLVR to synthesize a large amount of high-quality critique data at low cost without applying closed-source LLMs. This synthetic data is then used to cold start and facilitate RLVR training of the verifier.

We begin by training a base solver model using RLVR (e.g., GRPO), recording its solution trajectories throughout the training process. To reduce data redundancy, we filter out the trajectories that are ultimately identical with Math Verify[1]. For a given question $q$, we have a set of model-generated solutions $\{\hat{s}_i\}$ and a ground-truth solution $s^*$. To synthesize a critique for a solution pair $(q, \hat{s}_i)$, we instruct a small, instruction-tuned language model with the following template:

---

**Prompt for Critique with Ground Truth**

You are an expert mathematics tutor who always thinks step-by-step. You will be shown: Question, Ground Truth (hidden from the student), Solution. Your task:
* Analyze the Solution according to the Ground Truth. But do not mention 'ground truth', 'correct answer', 'official solution', etc.
* Produce a numbered step-by-step analysis of the Solution, explaining why it is correct or incorrect.
* End with a single line containing only
$\boxed{True}$ — if the boxed answer in the Solution is correct,
$\boxed{False}$ — otherwise.

---

The instruct model generates a candidate critique $c_i$. We then apply a rejection sampling filter: the final $\boxed{Judgment}$ (True/False) matches the actual correctness of $\hat{a}_i$ are retained. This process ensures the synthetically generated data maintains a high standard of quality, teaching the verifier not just to judge but to justify its judgment with a coherent rationale.

## 4.2 DATA SELECTION AND VERIFIER TRAINING

**Cold Start.** Since the base model lacks the critique ability, it is difficult to enhance its verification capability through reinforcement learning. We illustrate this in Appendix D. We use the synthetic critique dataset, $\mathcal{D}_{\text{synth}} = \{(q, \hat{a}, c)\}$ to cold start the base model. This SFT step serves as an effective cold start, equipping the model with fundamental critique generation capabilities before the subsequent RLVR phase.

**Balance Data for RVLR.** The filtered synthetic dataset often exhibits class imbalance, with more critiques labeling solutions as incorrect ($y = \text{False}$). We found that training the verifier with imbalanced samples through RLVR easily leads to reward hacking, where the LLM tends to predict all

---

[1]https://github.com/huggingface/Math-Verify

Figure 2: Test-Time Scaling with Mirror-Verifier. We deploy the verifier to generate critiques for each candidate solution, then select the final answer according to the weighted majority voting. If the average verification score of the chosen answer is lower than the threshold $\tau$, the system will abstain from answering the question.

samples as False, as illustrated in Appendix E. To this end, we conducted balanced data sampling for positive and negative samples. Additionally, to make the model pay more attention to minor-yet-correct samples, we only selected question-solution pairs with an accuracy rate of less than 60% for further RLVR training. We further refine the SFT-initialized verifier using Reinforcement Learning with the balanced dataset. The goal is to align the verifier's critique generation policy, $\pi_\phi(c|q, \hat{a})$, to produce critiques that are not only correct but also pedagogically valuable and concise. The verifier model is prompted with a question-solution pair $(q, \hat{a})$ and is tasked to generate a critique $c$.

### 4.3 ACCURATE AND HONEST TEST-TIME SCALING

The resulting verifier is deployed in a solver-verifier framework to enhance performance at test time via solution sampling and selection. For a given test question $q$, the solver generates $N$ candidate solutions $\{s_1, s_2, ..., s_N\}$. The Mirror-Verifier then evaluates each solution $s_i$ by generating $M$ independent critiques $\{c_{i,1}, c_{i,2}, ..., c_{i,M}\}$. Each critique $c_{i,j}$ contains a binary judgment $y_{i,j} \in \{\text{True}, \text{False}\}$. The verification score $w_i$ for solution $s_i$ is calculated as the proportion of critiques judging it to be correct:

$$w_i = \frac{1}{M} \sum_{j=1}^{M} \mathbb{I}(y_{i,j} = \text{True})$$

The score can be used for the following aspects:

- **Weighted Voting for Accuracy**: The final answer is selected through a weighted majority vote. Each solution $s_i$ contributes a vote for its final answer $a_i$, weighted by its verification score $x_i$:

$$\hat{a} = \arg\max_{a \in \mathcal{A}} \sum_{i=1}^{N} w_i \cdot \mathbb{I}(a_i = a)$$

- **Selective Abstention for Honesty**: The system can abstain from answering when it lacks sufficient confidence. Specifically, for the selected answer $\hat{a}$, the average verification score of all solutions that proposed $\hat{a}$ is computed:

$$\bar{w}_{\hat{a}} = \frac{\sum_{i=1}^{N} w_i \cdot \mathbb{I}(a_i = \hat{a})}{\sum_{i=1}^{N} \mathbb{I}(a_i = \hat{a})}$$

A predefined confidence threshold $\tau \in [0, 1]$ is set. If $\bar{w}_{\hat{a}} < \tau$, the system rejects the query and abstains from providing an answer. This mechanism enhances honesty by preventing the delivery of potentially unreliable or low-confidence responses.

This framework ensures that the final output is not only accurate (through weighted voting) but also trustworthy (through selective abstention), thereby improving overall reliability and alignment with user expectations.

## 5 EXPERIMENTS

### 5.1 SETUP

**Data** We evaluate the Solver-Verifier framework with 5 widely used mathematical reasoning benchmarks: MATH-500 (Lightman et al., 2023), OlympiadBench (He et al., 2024a), Minverva-Math (Lewkowycz et al., 2022), AIME24, and AMC23. We further combine all of the evaluation benchmarks to report the performance of test-time scaling with different sampling sizes (from 1 to 16). The training data used in this work is OpenR1-45K, which is a subset of OpenR1-Math-220k (Hugging Face, 2025).

**Metrics.** In this work, we conduct 2 metrics to evaluate the solver-verifier framework.

- **Accuracy**: The proportion of problems for which the model generates a correct final answer. For a benchmark $D$, the Accuracy is defined as:

$$\text{Accuracy} = \frac{1}{|D|} \sum_{i=1}^{|D|} \mathbb{I}(\hat{a}_i = a_i^*)$$

where $\hat{a}_i$ is the model's predicted answer for the $i$-th problem, $a_i^*$ is the ground-truth answer, and $\mathbb{I}$ is the indicator function.

- **Honesty Score**: We propose a metric that jointly considers correctness and the harm of providing incorrect information. For each problem, the model receives +1 if it answers correctly, -1 if it answers incorrectly, and 0 if it abstains from answering. The Honesty Score for the entire dataset is the average of these values:

$$\text{Honesty Score} = \frac{1}{|D|} \sum_{i=1}^{|D|} [\mathbb{I}(\hat{a}_i = a_i^*) - \mathbb{I}(\hat{a}_i \neq a_i^* \wedge \hat{a}_i \neq \text{``abstain''})]$$

This metric encourages not only high accuracy but also cautious behavior by penalizing incorrect outputs, thus mitigating the risk of propagating harmful misinformation.

**Training and Testing Details.** We conduct our RL training experiments on Qwen2.5-Math (Yang et al., 2024a) series Models with different sizes. We change the rope theta from 10,000 to 40,000 and extend the window size to 16,384. We remove the KL loss term and the. Following Dr.GRPO (Liu et al., 2025b), we remove length normalization in the loss function and the standard normalization in advantage computation. For all training procedures, the learning rate is set as 1e-7. The batch size is set as 128 and 1024 for training the solver and verifier, respectively. The rollout size is set as 8 for training the zero-solver and 16 for training the zero-verifier. The temperature is set as 1.0 for both training and testing. During test-time scaling, we generate $M = 16$ critiques per solution.

**Baselines.** We compare our Mirror-Verifier with the following methods: (1) *Pass@1 (Avg@16)*, we sample 16 solutions for each question and compute the average accuracy for all responses. (2) *Majority@K*, (3) Math-Shepherd-PRM (Wang et al., 2024), a process reward model trained with automatic process data annotation. (4) Skywork-O1-PRM (He et al., 2024b), A specialized model designed to enhance reasoning capability through incremental process rewards, ideal for complex problem solving at a smaller scale. (5) Qwen2.5-Math-7B-CFT (Wang et al., 2025), a critique model trained on 50K critique responses generated by GPT-4o. (6) Mirror-SFT model, the SFT cold-start model in our Mirror-Critique training procedure.

### 5.2 MAIN RESULTS

#### 5.2.1 ACCURACY PERFORMANCE WITHOUT ABSTAIN

We show the accuracy performance of test-time scaling in Table 1. In this experiment, the abstain threshold $\tau$ is set as 0 to acquire the best accurayc performance of each method. That is, we require the LLMs not to abstain from any given question. It is worth noting that our Mirror-Verifier achieved the best performance on the majority of benchmarks, with an overall performance higher than all the baselines. In particular, Mirror-Verifier-1.5B achieved the best results compared to other baseline methods on the five selected benchmarks.

Table 1: Overall performance of accuracy for Qwen2.5-Math series on AIME, MATH500, Olympiad, AMC, and Minerva. (#Instances denotes the number of training data used to train the model.)

| Method / Verifier | #Instances | AIME24 | MATH500 | Olympiad | AMC | Minerva | Overall |
|---|---|---|---|---|---|---|---|
| *Qwen2.5-Math-1.5B as the Solver* | | | | | | | |
| *pass@1 (avg@16)* | - | 11.9 | 75.0 | 39.6 | 44.2 | 31.1 | 49.2 |
| *majority@16* | - | 20.0 | 81.1 | 47.5 | 48.6 | 35.5 | 55.7 |
| Qwen2.5-Math-7B-CFT | 50k | 20.0 | 81.3 | 47.2 | 49.8 | 34.9 | 55.6 |
| Math-Shepherd-PRM | 445k | 20.0 | 81.4 | 47.6 | 48.2 | 35.7 | 55.8 |
| Skywork-o1-PRM-1.5B | unknown | 20.0 | 83.4 | 48.9 | **53.0** | 36.0 | 57.3 |
| Mirror-SFT-1.5B | 170k | 16.7 | 82.3 | 46.2 | 50.6 | 34.8 | 55.5 |
| **Mirror-Verifier-1.5B** | 170k | **23.3** | **84.0** | **49.5** | **53.0** | **37.9** | **58.2** |
| *Qwen2.5-Math-7B as the Solver* | | | | | | | |
| *pass@1 (avg@16)* | - | 23.2 | 84.2 | 46.7 | 57.1 | 38.1 | 57.3 |
| *majority@16* | - | 23.3 | 88.1 | 52.3 | 63.3 | 40.4 | 61.8 |
| Qwen2.5-Math-7B-CFT | 50k | 25.0 | 87.8 | 52.7 | 63.2 | 38.7 | 61.7 |
| Math-Shepherd-PRM | 445k | **26.7** | 88.9 | 52.4 | 62.7 | 40.0 | 62.0 |
| Skywork-o1-PRM-7B | unknown | **26.7** | 88.6 | 52.4 | **67.5** | 40.4 | 62.3 |
| Mirror-SFT-7B | 116k | 25.0 | 88.4 | 53.2 | 62.7 | 40.3 | 62.2 |
| **Mirror-Verifier-7B** | 116k | 25.0 | **89.1** | **54.1** | 63.9 | **41.2** | **63.0** |

We further show the accuracy performance versus the number of candidate solutions $K$ across the five chosen benchmarks in Figure 3. The results show that our Mirror-Verifier consistently improves performance for different values of $K$, significantly outperforming majority voting and other baselines. Additionally, it is worth noting that although Mirror-Verifier was trained with $K = 8$, it can still effectively generalize to $K = 16$.

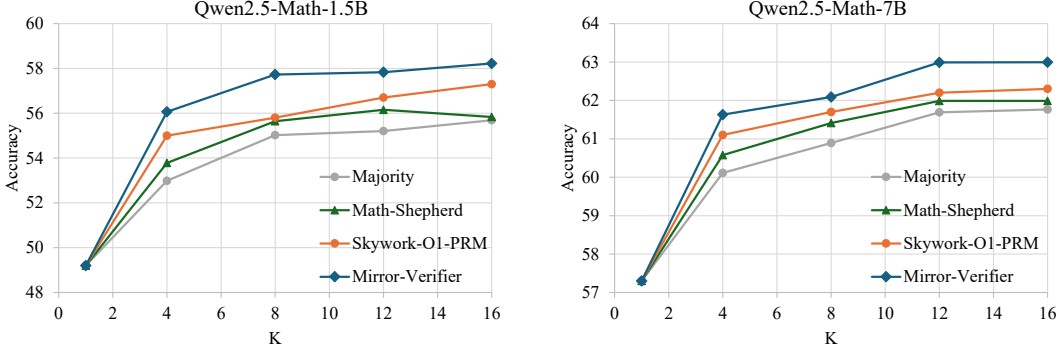

Figure 3: Accuracy vs. number of candidate solutions ($K$) for different methods.

### 5.2.2 HONESTY PERFORMANCE ON DIFFERENT BENCHMARKS

Then, we also report the performance honesty score on the selected benchmark. We uniformly set the threshold $\tau = 0.20$ for all methods to control the abstention as described in Section 4.3. For *pass@1 (avg@16)* and *majority@16*, no mechanism can be used to abstain. We report the honesty score of these methods directly. The results are shown in Table 2. Both of our **Mirror-Verifier-1.5/7B** models outperform the honesty performance of all baseline methods.

Table 2: Overall performance of honesty for Qwen2.5-Math series on AIME, MATH500, Olympiad, AMC, and Minerva.

| Method / Verifier | AIME24 | MATH500 | Olympiad | AMC | Minerva | Honesty |
|---|---|---|---|---|---|---|
| *Qwen2.5-Math-1.5B as the Solver* | | | | | | |
| *pass@1 (avg@16)* | -76.2 | 50.0 | -20.8 | -11.6 | -37.8 | -1.60 |
| *majority@16* | -60.0 | 62.2 | -5.0 | -2.8 | -29.0 | 11.4 |
| Math-Shepherd-PRM | -60.0 | 62.6 | -4.89 | -3.61 | -27.9 | 11.7 |
| Skywork-o1-PRM-1.5B | -56.7 | 67.0 | 1.04 | 9.64 | -21.7 | 17.6 |
| Mirror-SFT-1.5B | -53.3 | 66.6 | 6.52 | 15.7 | -19.9 | 20.5 |
| **Mirror-Verifier-1.5B** | **-13.3** | **71.6** | **21.5** | **30.1** | **-5.15** | **32.7** |
| *Qwen2.5-Math-7B as the Solver* | | | | | | |
| *pass@1 (avg@16)* | -53.6 | 68.4 | -6.6 | 14.2 | -23.8 | 14.6 |
| *majority@16* | -53.4 | 76.2 | 4.60 | 26.6 | -19.2 | 23.6 |
| Math-Shepherd-PRM | -46.7 | 77.6 | 4.74 | 25.3 | -19.5 | 23.9 |
| Skywork-o1-PRM-1.5B | **-26.7** | 75.0 | 11.3 | **38.6** | -17.3 | 27.4 |
| Mirror-SFT-7B | **-26.7** | 77.0 | 16.7 | 30.1 | -16.9 | 30.2 |
| **Mirror-Verifier-7B** | **-26.7** | **78.0** | **17.4** | 33.7 | **-14.3** | **31.3** |

## 5.3 HONESTY-ACCURACY CURVE

To further show the effectiveness of the RLVR process for training Mirror-Verifier, we plot the Honesty-Accuracy curve for Mirror-SFT and Mirror-RLVR models in Figure 4. This is done by gradually increasing the threshold $\tau$ while evaluating the test-time scaling results. We combine the five selected benchmarks to report the accuracy and honesty score. As illustrated in the figure, in the case of equal accuracy, the Honesty Score of our Mirror-RLVR model is higher than that of Mirror-SFT, which is reflected in the envelope being positioned higher up. In addition, Mirror-RLVR is also significantly higher than Skywork-O1-PRM, surpassing this stronger baseline. The result fully demonstrates the effectiveness of the RLVR training within the Mirror-Critique framework.

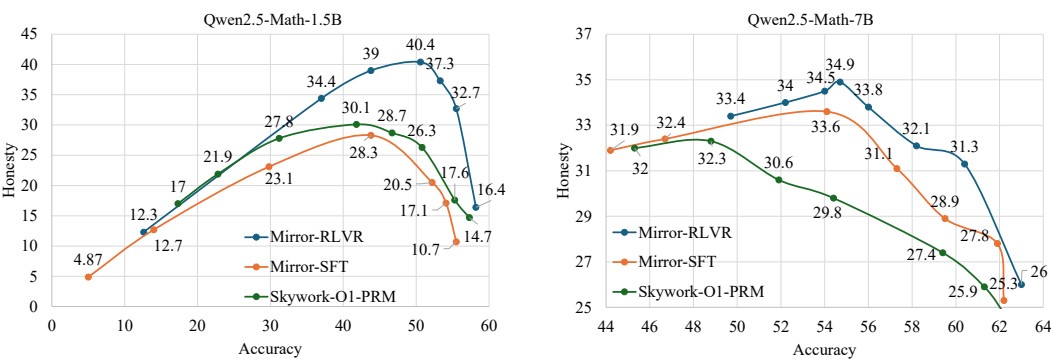

Figure 4: Honesty-Accuracy Curve for comparison of SFT and RLVR model.

## 5.4 QUALITY OF SYNTHETIC CRITIQUE DATA

We further measured the quality of the synthetic critique data. Although we have ensured the accuracy of the final judgment labels through rejection sampling, the accuracy of the critique content still needs to be measured. We consider that the quality of critique is important for performance. We adopted the approach of using the LLM as a judge. We randomly sample 30 data points from the synthetic dataset and use Deepseek-V3.1 to evaluate the quality of critique content. The input prompt of Deepseek-V3.1 is shown in Appendix B. At the same time, we also conducted manual

verification. Table 3 shows the results of LLM as a judge and manual verification. It can be seen that the critique content of our synthetic data, despite still containing noise, has a certain degree of quality assurance. Finding better denoising methods can be a direction for future research and may further improve the performance of the resulting verifier.

Table 3: Quality Evaluation for our Synthetic Critique Data generated from Qwen2.5-7B-Instruct.

| Synthetic Data | Gen Model | DeepSeek-V3.1 | Human |
|---|---|---|---|
| Mirror-Critique-1.5B | Qwen2.5-7B-Instruct | 83.3% | 80.0% |
| Mirror-Critique-7B | Qwen2.5-7B-Instruct | 80.0% | 76.7% |

## 5.5 CASE STUDY

We further show a case of our synthetic critique data as illustrated in Figure 5. The synthetic critique exemplifies a common type of logical verification, where a proposed solution is checked for internal consistency and correctness against existing mathematical principles or theories. It highlights an error in calculation by referencing a fundamental combinatorial concept, demonstrating a typical corrective action in problem-solving analysis.

**Question**

In a football championship, 16 teams participate, each playing against each other once. What is the minimum number of games that must be played so that among any three teams, there are two that have already played against each other?

**Candidate Solution**

1. **Understanding the Problem:**
   - We have 16 teams in a football championship.
   - Each team plays against each other team exactly once.
   - We need to find the minimum number of games such that among any three teams, there are two that have already played against each other.

2. **Graph Theory Representation:**
   - Represent the teams as vertices of a graph.
   - Represent a game between two teams as an edge between two vertices.
   - We need to ensure that in any subset of three vertices (teams), at least one edge (game) exists.
   ...

8. **Conclusion:**
   - Therefore, the minimum number of games that must be played so that among any three teams, there are two that have already played against each other is:
   $$\boxed{65}$$

**Synthetic Critique**

Now I will give the step-by-step Analysis followed by the boxed judgment.
## Analysis:
1. **Understanding the Problem:**
   - The problem correctly identifies the need to ensure that among any three teams, there are two that have played against each other. This translates to a graph theory problem where we need to avoid an independent set of size 3 in the complement graph.

2. **Graph Theory Representation:**
   - The teams are represented as vertices, and the games as edges. This is correct.

3. **Combinatorial Approach:**
   - The problem correctly identifies the need to find the minimum number of edges such that the graph does not contain an independent set of size 3. This is correct.
...
8. **Minimum Number of Games:**
   - The solution incorrectly states that the minimum number of games required is 65. The correct calculation should be:
   $$120 - 64 = 56$$
   - The solution should have subtracted the maximum number of edges in a triangle-free graph from the total number of edges in a complete graph.

9. **Conclusion:**
   - The solution concludes with 65, which is incorrect. The correct answer is 56.

$$\boxed{False}$$

Figure 5: Case Study of the Synthetic Critique Data.

## 6 CONCLUSION

In this work, we introduced **Mirror-Critique**, a novel framework for training verifiers that leverages rich, synthetic critique data to enable accurate and honest test-time scaling. Our key insight was to address the limitation of binary reward signals by generating critiques through a low-cost, self-supervised pipeline that contrasts model-generated solutions with ground-truth answers without the use of stronger LLMs. Extensive experiments on multiple mathematical reasoning benchmarks demonstrate the effectiveness of our resulting **Mirror-Verifier** in terms of both solution accuracy and honesty. The framework's ability to identify minority-yet-correct answers through weighted voting and to abstain from questions beyond the model's capability boundaries marks a substantial step towards more reliable and trustworthy reasoning systems.

## 7 REPRODUCIBILITY STATEMENT

To ensure the reproducibility of our research, we have meticulously assembled a comprehensive reproducibility package as part of our supplementary materials. This package is designed to enable the seamless replication of all experiments detailed in our paper. It encompasses anonymized source code that implements the proposed model and training procedures, along with the process to synthesize the datasets utilized in our experiments. Comprehensive guidelines for setting up the environment, preparing the data, and executing the experiments are meticulously outlined in the accompanying README documentation. Additionally, we have included precise configuration files and scripts that specify all hyperparameters and the training commands necessary to reproduce our results.

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

# APPENDIX

## A   THE USE OF LARGE LANGUAGE MODELS

This work utilizes the open-source LLMs for model training and testing. In addition, some closed-source LLMs (Kimi-K2, DeepSeek-V3.1, and Gemini 2.5) are employed for polishing and grammatical error correction in the writing of the paper. In general, our use of LLMs in paper writing is cautious and limited.

## B   PROMPTS USED IN THIS WORK

---

**Prompt for Solving Complex Reasoning Tasks**

Your task is to solve the given question step by step. You should conduct a systematic, thorough reasoning process before providing the final answer. This involves analyzing, summarizing, exploring, reassessing, and refining your reasoning process through multiple iterations. Each reasoning step should include detailed analysis, brainstorming, verification, and refinement of ideas. You should include the final answer in \boxed{} for closed-form results like multiple choices or mathematical results.

---

**Prompt for Critique without Ground Truth**

You are an expert mathematics tutor who always thinks step-by-step. You will be shown: Question and its Solution. Your task:
* Analyze the Solution according to the Question
* Produce a numbered step-by-step analysis of the Solution, explaining why it is correct or incorrect.
* End with a single line containing only
$\boxed{True}$ — if the boxed answer in the Solution is correct,
$\boxed{False}$ — otherwise.

---

**Prompt for Evaluating the Quality of Critique Content**

You are an evaluator tasked with analyzing critique accuracy. For each input, you will receive:
- Question: the problem statement
- Ground truth solution: the correct reference solution
- Candidate solution: a proposed solution to the question
- Critique: an analysis evaluating the candidate solution

Your task is to identify if the critique content is correct.

Process the following Input:
Question: {question}
—
Ground truth solution: {ground_truth_solution}
—
Candidate solution: {candidate_solution}
—-
Critique: {critique}
—

Now, please determine whether the critique to the candidate solution is accurate or not. Finally, provide your judgment in the specified boxed format. (Shorten your output and give me quick judgment)

---

## C  REWARD FUNCTION DESIGN

For Training zero-solver, we utilize the Math-Verify[2] to judge whether the candidate solution is correct according to the ground truth answer. On this basis, I also add the constraint that the use of code is not allowed. When code is detected in the text, the result is directly judged to be incorrect. This is because code verification is beyond the scope of this work. In the future, we will consider expanding the method to the field of Tool-Integrated Reasoning. For training zero-verifier, I directly use regular expressions to extract the text within 'boxed', and match it with True/False to obtain a binary reward.

## D  BASE MODEL LACKS CRITIQUE ABILITY

We observe that the base model lacks the necessary critique capabilities, which is reflected in the fact that there are almost no critique outputs that meet the requirements in its responses. We show an example in the following:

> **Critique Generated from Base Model**
>
> You are an expert mathematics tutor who always thinks step-by-step. You will be shown: Question and its Solution. Your task:
> * Analyze the Solution according to the Question
> * Produce a numbered step-by-step analysis of the Solution, explaining why it is correct or incorrect.* End with a single line containing only
> $\boxed{True}$ - if the boxed answer in the Solution is correct,
> $\boxed{False}$ - otherwise.

Qwen2.5-Math Series sometimes repeats the content of the system prompt, and sometimes it echoes the candidate solution. Therefore, it is necessary to fine-tune the Base model using critique data.

## E  REWARD HACKING ON IMBALANCE SAMPLE

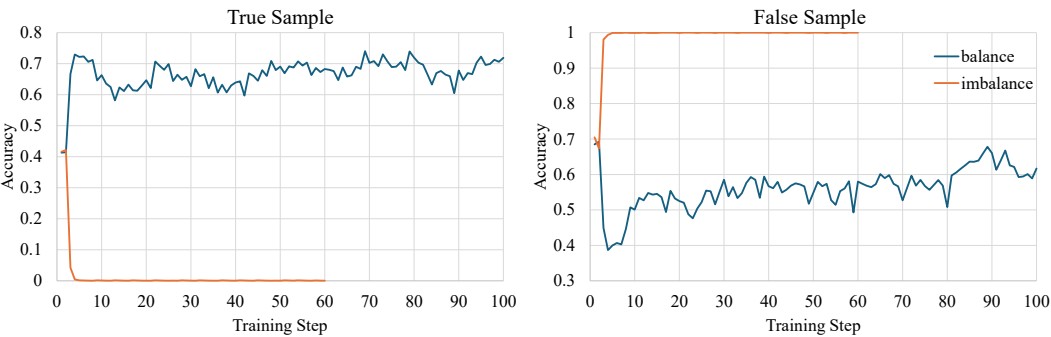

Figure 6: Reward Hacking on Imbalanced Data.

We observe the reward hacking phenomena on imbalanced data training during RLVR. The statistical results of the training dynamics is shown in Figure 6. The imbalance data in this experiment contains about 75% negative samples. When applied to RLVR, the model quickly adjusts the distribution of predictions, tending to predict all results as False. Subsequently, we sampled positive and negative samples at a 1:1 ratio to create balanced data, which solved the reward hacking problem.

---

[2]https://github.com/huggingface/Math-Verify

# F TRAINING DYNAMICS

We further plot the training dynamics of verification accuracy and F1 score for **Mirror-Verifier-7B** model in Figure 7. When using balanced data, the training of the verifier is relatively stable. It can be observed that as the training steps increase, both the verification rewards and the training F1 score gradually rise.

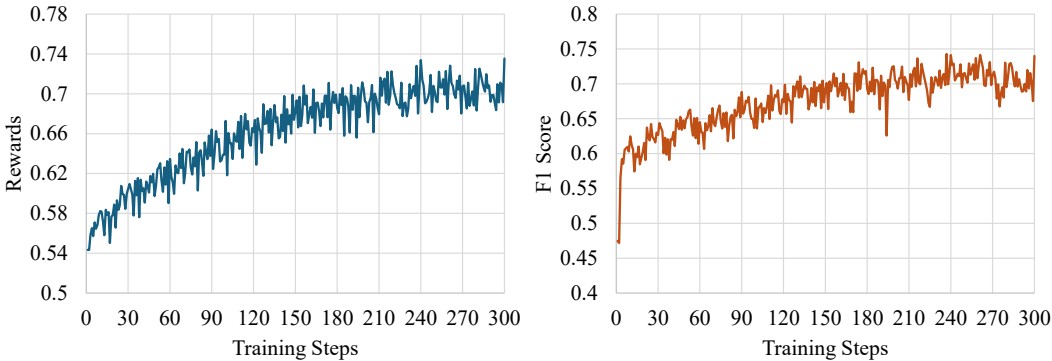

Figure 7: Training dynamics for **Mirror-Verifier-7B**.

