# OpenReview forum: "Critique to Verify: Accurate and Honest Test-Time Scaling with RL-Trained Verifiers"
_ICLR.cc/2026/Conference — ICLR 2026 Conference Withdrawn Submission_

### Official Review · Reviewer_U4fi · 2025-10-16

**Soundness:** 3
**Presentation:** 2
**Contribution:** 3
**Rating:** 4
**Confidence:** 4

**Summary:**

This paper introduces Mirror-Critique, a framework for training verifiers that enhance test-time scaling in Large Language Models (LLMs) for mathematical reasoning tasks. The key innovation is the synthesis of high-quality critique data by contrasting model-generated solutions with ground-truth answers, using rejection sampling on a small instruction-tuned model. These critiques are then used to cold-start and train a verifier via Reinforcement Learning with Verifiable Reward (RLVR). At test time, the Mirror-Verifier generates multiple critiques per solution candidate, aggregating them into verification scores used for weighted voting and selective abstention. The authors evaluate their approach on five mathematical reasoning benchmarks (MATH-500, OlympiadBench, Minerva-Math, AIME24, AMC23) and introduce a "honesty score" metric that jointly considers correctness and appropriate abstention behavior.

**Strengths:**

- The method of generating critique data by contrasting model solutions with ground truth using rejection sampling from open-source models is creative and cost-effective, avoiding dependence on closed-source APIs (Section 4.1).
- The paper evaluates on five diverse mathematical benchmarks (MATH-500, OlympiadBench, Minerva-Math, AIME24, AMC23) with multiple model sizes, demonstrating consistent improvements (Table 1).
- The honesty score is a valuable contribution that jointly considers correctness and abstention, addressing an important but understudied aspect of LLM reliability (Section 5.1).
- Mirror-Verifier shows consistent improvements over strong baselines including Math-Shepherd-PRM and Skywork-O1-PRM across most benchmarks, particularly for the 1.5B model (Table 1).
- The paper provides useful ablations showing the importance of balanced data sampling (Appendix E) and the effectiveness of RLVR training over SFT alone (Figure 4).

**Weaknesses:**

- While the overall framework is coherent, individual components (RLVR training, critique generation, weighted voting) are relatively standard. The main contribution is their combination rather than fundamental algorithmic innovation.
- Synthetic data quality concerns: The critique data quality evaluation (Table 3) shows only 76.7-80% accuracy by human evaluation, indicating substantial noise. The paper does not adequately address how this noise impacts final performance or explore denoising methods.
- The paper requires generating M=16 critiques per solution during inference, which could be computationally expensive. No analysis of computational costs, inference time, or comparison with baseline methods is provided.
- The paper lacks theoretical analysis of why contrasting with ground truth produces better critiques than other approaches, or when the weighted voting strategy should outperform alternatives.
- The paper mentions concurrent work on solution aggregators (SSA, AggLM) but does not provide empirical comparisons. This makes it difficult to assess the relative merits of the proposed approach.
- While Figure 4 shows Pareto curves, the paper does not provide guidance on how to set the threshold τ in practice or analyze the tradeoff systematically across different difficulty levels or problem types.
- The paper contains some grammatical errors and unclear statements (e.g., "We remove the KL loss term and the." in Section 5.1 appears incomplete). The organization could be improved—for instance, the reward function design is relegated to Appendix C but is crucial for understanding the method.

**Questions:**

- What is the computational overhead of generating 16 critiques per solution compared to baseline methods? Can you provide wall-clock time comparisons and discuss the practical feasibility for real-world deployment?
- How does the quality of synthetic critique data (76.7-80% accuracy) affect final performance? Have you experimented with filtering or weighting critiques based on confidence, or with iterative refinement of the critique model?
- Can this approach generalize to other domains where ground-truth solutions may not be readily available or verifiable (e.g., open-ended reasoning, creative writing)?
- How should practitioners set the abstention threshold τ in practice? Can you provide principled guidelines or an automated method for threshold selection based on desired accuracy-coverage tradeoffs?

---

### Official Review · Reviewer_ctKo · 2025-11-03

**Soundness:** 3
**Presentation:** 2
**Contribution:** 2
**Rating:** 4
**Confidence:** 3

**Summary:**

This paper focuses on improving verifiers for test-time scaling, since existing reward models still struggle to select the correct answer among candidate generations and in fact may still perform worse than majority voting. Most existing verifiers are trained using binary signals; instead, this paper aims to train verifiers using more informative critique data. To tackle the problem of creating high-quality critique data cheaply, the paper uses RLVR on a base model to obtain trajectories, which are then compared and critiqued against ground-truth solutions using a small instruction-tuned model. This critique dataset is then used in both SFT and RL to train the mirror verifier. Experiments show that scaling test-time compute and verifying generations using the mirror verifier outperforms using other verifiers that are trained with synthetic data.

**Strengths:**

Quality:
- The paper provides analysis of the quality of the synthetic critique data.
- Empirical results demonstrate that both mirror verifier 1.5B and 7B outperform other verifiers of their size on average.

Significance:
- The paper demonstrates that we can create high-quality critique data and improve verification by SFT/RLing on it instead of binary rewards.

**Weaknesses:**

Originality
- The framework combines well-known components (RL, synthetic data) in a fairly standard way; the novelty of the overall approach is unclear.

Quality
- Would be interested to see more ablations, such as using binary signal on OpenR1-45K instead of critique; how much of the gains over other verifiers come from simply using the OpenR1 dataset?
- No clear measure of the oracle rate (e.g., existence of a correct answer within the K samples), or how closely the mirror verifier approaches it.
- Would benefit from stronger experimental analysis: e.g., scaling beyond 16 test-time samples and ablations over M (number of critiques per solution).

Clarity
- In line 186, it is unclear what is the "small, instruction-tuned language model"used for synthesizing a critique. Is it the solver model or a different LLM? I could not find what was used for this model in the experiments in section 5. This makes it difficult for me to understand if the method is more of a self-contained, self-improvement approach, or an approach for producing a good verifier that can applied everywhere.
   - If it is a self-improvement approach, I would be curious about self-verification baselines (https://arxiv.org/abs/2502.01839)
   - If the aim is to produce a good verifier, I wish to see more evaluation, such as table 1 on other solver models.
- The presentation of the problem setup and pipeline is confusing, making it hard to follow what is learned at each stage.

**Questions:**

(summarized from weaknesses above)
1. How would a verifier trained on binary signal on OpenR1-45K perform?
2. What is the oracle rate for the results in Table 1?
3. How does mirror verifier behave with different K and M?
4. Can you clarify what the "small, instruction-tuned language model" is in line 186?
5. How does a self-verification baseline perform? How about using the mirror verifier to scale test time compute on other solver models?

---

### Official Review · Reviewer_UxrR · 2025-11-03

**Soundness:** 1
**Presentation:** 1
**Contribution:** 2
**Rating:** 0
**Confidence:** 3

**Summary:**

This paper presents Mirror-Critique, a SFT-RLVR combined approach to create a verifier for test-time-scaling (weighted majority voting). First, this paper proposes a synthetic data generation approach by (1) prompting an instruct-model to generate critiques from the training step of zero-solver; and (2) running rejection sampling as cold-start data for Mirror-SFT. Second, the cold-start model is then trained with RLVR to further improve the critique-based verifier's performance. Further, the authors propose a novel metric, honesty score, to further quantify the robustness of the "confidence" assigned by the verifier ensembling method.

**Strengths:**

1. this work applies the recent RLVR approach into a new application, verifier training, and show its promises.

2. this work proposes honesty scores, and shows that proposed approach is more reliable than baselines on abstention on wrong / low-confidence responses.

**Weaknesses:**

Although the concept is interesting, the clarity (as detailed in the questions part), rigorosity, and significance is rather limited.

1. this paper mentioned zero-solver training before the verifier training, but it seems that in Table 1 only vanilla llm is used as solver. Is the zero-solver training only used for generating synthetic data? Further, if the focus of this work is to push the LLM's capability on math reasoning, it should be compared against SOTA RLVR trained models, e.g. deepscaler-1.5b [1] and R1-distilled-1.5B model[2], which seems to be much stronger than the proposed approach.

2. the authors should present the computational cost analysis (e.g. total token generation or flops) of the proposed approach, and compare with different baselines, and adding the aforementioned verifier-free approach, deepscaler-1.5b [1] and R1-distilled-1.5B model[2] etc.. My understanding is that mirror-critique is very expensive and not practical.

3. the authors used openR1-45k dataset, which is an extension on AI-MO/NuminaMath-1.5[3], that covers aime etc. and potentially overlaps with evaluations. the authors should describe their efforts on data contamination analysis.

4. the authors should also add ablation studies on RVLR training (1) without SFT cold-start (2) without CoT critique, i.e. RLVR only on the correctness labels.

5. analysis needs to be added on why proposed approach is better than baselines on "honesty scores".


[1] https://huggingface.co/agentica-org/DeepScaleR-1.5B-Preview

[2] https://huggingface.co/deepseek-ai/DeepSeek-R1-Distill-Qwen-1.5B

[3] https://huggingface.co/datasets/AI-MO/NuminaMath-1.5

**Questions:**

1. in the third step: RLVR verifier after cold-start, how is critique verifiable? is it just using the binary category?

2. why do the authors use verifier ensembling approach, instead of using the token probabilities of yes/no on solution correctness? it seems very expensive and not practical.

clarity questions and comments:

* line 54: “Base LLM” should be “base LLM”

* line 87-88: “zero-RL” where does this definition come from? the cited reference seems not touching this concept.

* line 147-148: “four components” but only (1-3)?

* line 213: should “y=False” be “a=False”?

* line 301: “We remove the KL loss term and the.” something missing at the end?

* line 320: “accurayc” should be “accuracy”

* line 395: should it be “*Skywork-o1-PRM-7B*” model?

* table 1: what’s the size of the training data, why are 1.5b model and 7b reported to be trained on different data size. and further, line 278 mentioned OpenR1-45K, which is different from the reported data sizes in table 1. this is very confusing.

* last but not least: maybe i missed this, is Figure 2 ever referred in the main text?

---

### Official Review · Reviewer_iSoz · 2025-11-08

**Soundness:** 3
**Presentation:** 3
**Contribution:** 2
**Rating:** 4
**Confidence:** 4

**Summary:**

This paper introduces Mirror-Critique, a framework for training verifier models to improve test-time reasoning performance in LLMs. The authors posit that common aggregation methods, such as majority voting, are limited in their ability to identify correct solutions that appear in the minority. This limitation is attributed to verifiers being trained on simple binary feedback rather than more informative critique signals.

The proposed method trains a "Mirror-Verifier" using synthetic critiques. This data is generated by contrasting model-generated solutions with ground-truth solutions, a process intended to teach the verifier why a solution is incorrect. The resulting verifier is deployed at test time to:
- Aggregate solutions using a weighted majority vote based on verification scores.
- Selectively abstain from answering if the average verification score for a chosen answer is below a set threshold, which is designed to improve model honesty.

**Strengths:**

- The paper's core strength is its "Mirroring the Truth" technique for synthesizing critiques. Instead of relying on expensive human data or proprietary models, this method cleverly contrasts model-generated solutions with ground-truth answers to create a rich, instructive dataset. This "self-contained" approach  for teaching a verifier why a solution is wrong, not just if it is wrong, is a highly original contribution.
- The paper provides a significant contribution by formally defining and evaluating model "honesty". The introduction of the "Honesty Score" metric and the corresponding "Selective Abstention" mechanism  moves beyond simple accuracy. This provides a clear, practical framework for optimizing and measuring a model's ability to recognize its own limitations, which is a critical step for building more trustworthy AI systems.

**Weaknesses:**

1. Insufficient Analysis of the Core Motivation ("Minority-Yet-Correct" Solutions): The paper is well-motivated by the claim that existing methods fail to identify "minority-yet-correct" solutions. However, the experiments do not provide a direct analysis to confirm that the Mirror-Verifier is specifically solving this problem. While Table 1 shows strong overall accuracy, it's impossible to tell if this gain comes from better-weighted voting in general or from successfully recovering these specific minority solutions. To substantiate this core claim, the paper would be significantly strengthened by an analysis that:

  - Identifies the set of problems in the test data where the correct answer is a minority answer (e.g., appears in < 50% of the $N$ samples).
  - Compares the accuracy of Mirror-Verifier, majority voting, and other baselines specifically on this subset of problems.

2. Incomplete Baseline Comparison for the "Honesty" Metric: The "Honesty Score" is a valuable contribution for evaluating model reliability. However, the evaluation in Table 2 is potentially unfair. It applies a single, uniform abstention threshold ($\tau=0.20$) to all verifier-based methods. Different models (e.g., Mirror-Verifier, Skywork-01-PRM) produce verification scores on different scales; their scores are not calibrated with each other. A single threshold will arbitrarily favor whichever model happens to have its optimal trade-off point near $\tau=0.20$. A more rigorous comparison, which is already partially implemented in Figure 4, would be to plot the full Honesty-Accuracy curve for all verifier baselines (not just Mirror-SFT and Skywork-01-PRM ). This would allow for a comparison of the Pareto frontiers, showing which method achieves a better honesty-to-accuracy trade-off across all possible thresholds.

3. Unexplored Impact of Synthetic Data Noise: The paper's own quality evaluation (Table 3) shows that the synthetic critiques, while good, are not perfect, with human-rated accuracy around 76.7-80.0%. This means the verifier is being trained on a dataset with ~20% noisy labels (i.e., flawed critiques). The paper acknowledges this as an area for future work  but does not analyze the downstream impact of this noise in the current experiments.

**Questions:**

- Fairness of Honesty Score: Using a single abstention threshold ($\tau=0.20$) in Table 2 seems unfair, as models are not calibrated to the same scale. Could you justify this choice? To make the comparison fair, could you please provide the full Honesty-Accuracy curves (like in Figure 4) for all verifier baselines?

- "Self-Contained" Claim: You state the method is "self-contained" and avoids "stronger LLMs". However, the 7B verifier was trained on data from a 7B instruct model. Could you please clarify this claim? How should we view the novelty when a 7B model is trained on data from another model (even it is the same size)?

---

### Note · Authors · 2025-11-21

I have read and agree with the venue's withdrawal policy on behalf of myself and my co-authors.